# Administration of Bovine Milk Oligosaccharide to Weaning Gnotobiotic Mice Inoculated with a Simplified Infant Type Microbiota

**DOI:** 10.3390/microorganisms9051003

**Published:** 2021-05-06

**Authors:** Louise Margrethe Arildsen Jakobsen, Ulrik Kræmer Sundekilde, Henrik Jørgen Andersen, Witold Kot, Josue Leonardo Castro Mejia, Dennis Sandris Nielsen, Axel Kornerup Hansen, Hanne Christine Bertram

**Affiliations:** 1Department of Food Science, Aarhus University, Agro Food Park 48, 8200 Aarhus N, Denmark; uksundekilde@food.au.dk (U.K.S.); hannec.bertram@food.au.dk (H.C.B.); 2Arla Food Ingredients P/S, Sønderhøj 10, 8260 Viby J, Denmark; hejan@arlafoods.com; 3Department of Plant and Environmental Sciences, University of Copenhagen, Thorvaldsensvej 40, 1871 Frederiksberg C, Denmark; wk@plen.ku.dk; 4Department of Food Science, University of Copenhagen, Rolighedsvej 30, 1958 Frederiksberg C, Denmark; jcame@food.ku.dk (J.L.C.M.); dn@food.ku.dk (D.S.N.); 5Department of Veterinary and Animal Sciences, Faculty of Health and Medical Sciences, University of Copenhagen, Grønnegårdsvej 17, 1870 Frederiksberg C, Denmark; akh@sund.ku.dk

**Keywords:** infant nutrition, microbiome, NMR metabolomics, sialylated oligosaccharides, synthetic gut microbiota

## Abstract

Bovine milk oligosaccharides (BMO) share structural similarity to selected human milk oligosaccharides, which are natural prebiotics for infants. Thus, there is a potential in including BMOs as a prebiotic in infant formula. To examine the in vivo effect of BMO-supplementation on the infant gut microbiota, a BMO-rich diet (2% *w*/*w*) was fed to gnotobiotic mice (*n* = 11) inoculated with an infant type co-culture and compared with gnotobiotic mice receiving a control diet (*n* = 9). Nuclear magnetic resonance metabolomics in combination with high-throughput 16S rRNA gene amplicon sequencing was used to compare metabolic activity and microbiota composition in different compartments of the lower gastrointestinal tract. BMO components were detected in cecum and colon contents, revealing that BMO was available for the gut bacteria. The gut microbiota was dominated by *Enterobacteriaceae* and minor abundance of *Lactobacilliaceae*, while colonization of *Bifidobacteriaceae* did not succeed. Apart from a lower *E. coli* population in cecum content and lower formate (in colon) and succinate (in colon and cecum) concentrations, BMO supplementation did not result in significant changes in microbiota composition nor metabolic activity. The present study corroborates the importance of the presence of bifidobacteria for obtaining microbial-derived effects of milk oligosaccharides in the gastrointestinal tract.

## 1. Introduction

Human milk contains nutritional components for the growing infant and the microbiota of the infant. Around 15% of the energy in human breast milk is composed of complex carbohydrates, the so-called human milk oligosaccharides (HMOs). HMOs are nutritionally unavailable to the infant; however, they act as prebiotics that are either selectively metabolized by gut bacteria, whereby they become vital for establishment of a healthy microbiota in the infant [1], or they provide protection against infections through decoy of pathogens [2,3]. The first 1000 days (from conception until 2 years of age) of a child’s life is an important period in regard to establishing the gut microbiota and is particularly important for the healthy development of the child [4]. During weaning (6–24 months postpartum), the child transitions from exclusive breast-feeding towards gradual introduction of solid foods. It appears that breast milk maintains a *Bifidobacterium*-rich gut microbiota as long as the child is partially breastfed, despite introduction of solid foods [4]. Infant formula is sometimes enriched with galacto-oligosaccharides to mimic some of the bifidogenic effects of oligosaccharides in human milk. Additionally, studies have shown beneficial effects of synthetic galacto-oligosaccharides and lactose in modulating the expression of mucus-related genes and ascribing this to a structure specific effect [5]. Another group of oligosaccharides in human colostrum and mature milk are the sialylated milk oligosaccharides, with 6′-sialyllactose and 3′-sialyllactose being the most abundant (up to 500 and 300 mg/L, respectively) [6]. A higher content of sialylated oligosaccharides in breast milk from Malawian mothers appears to correlate positively with the growth of severely stunted infants and mouse and piglet studies have confirmed this effect to be mediated through an improved microbiota-dependent utilization of nutrients in the Malawian diets [7]. Bovine milk oligosaccharides (BMOs) represent a good source of sialylated oligosaccharides [8]. Since sialylated oligosaccharides are only present in trace amounts in infant formula (14–288 mg/L) [6], there is a potential for improving the bifidogenic effect of formula milk through enrichment with sialylated oligosaccharides. In a previous in vitro study, we found that BMO positively affects the composition of an infant type co-culture [9]. The infant type co-culture consisted of a total of eight bacteria representative of the infant gut microbiota, including strains of *Bifidobacterium*, *Lactobacillus*, *Clostridium*, *Escherichia*, *Parabacteroides* and *Staphylococcus*. The co-culture studies showed that BMO in combination with lactose or galactooligosaccharides resulted in a metabolic pattern characterized by low pH and high content of organic acids, particularly lactate. Furthermore, a combination of BMO and lactose proved to support growth of *Bifidobacterium longum* subsp. *longum*, while at the same time inhibiting growth of *C. perfringens* and *E. coli* [9]. Thus, there appears to be a potential benefit in introducing a bifidogenic effect by including BMO as a prebiotic ingredient in infant formula. The aim of the present study was to inoculate germ-free mice with an infant type co-culture to test the in vivo effect of a BMO-rich diet (2% *w/w*) on the composition and metabolic activity of the microbiota.

## 2. Materials and Methods

### 2.1. Experimental Diets

The experimental diets were formulated by Research Diets (Research Diets, Inc. New Brunswick, NJ, USA). The BMO diet (BMO) contained bovine milk oligosaccharide enriched whey (Lacprodan SAL-10^®^, Arla Foods Ingredients Group P/S, Aarhus, Denmark) composed of 68% (*w/w*) lactose and 10% (*w/w*) oligosaccharides, including 4.5% sialyllactose, as well as other acidic and neutral bovine milk oligosaccharides (not quantified). The control diet (CON) contained no BMO and the lactose content was controlled for by addition of a similar amount as present in the BMO diet (14/100 g). The nutritional composition of the two diets is given in Table 1.

The diets were formulated with isocaloric macronutrient compositions: protein (18%), carbohydrate (66%) and fat (16%). The source of fibers in the BMO diet was oligosaccharides (2.0/100 g) and cellulose (6.8/100 g), whereas in the CON diet the fibers purely consisted of cellulose (9.3/100 g).

The synthetic diets were packed in 500 g portions in heat-sealed bags under continuous nitrogen purge and the bags were subsequently irradiated twice.

### 2.2. Animals

Germ-free C57BL/6NTac mice (Taconic Biosciences, Ll. Skensved, Denmark) (*n* = 26) were raised with the mother in a sterile environment until three weeks after birth. The mice were then moved to a separate isolator where they were weaned on a standard isolator irradiated diet (Altromin1314, Brogaarden, Lynge, Denmark) for 1 week until inoculation with the 8-strain infant type co-culture. On the day of inoculation, the mice were randomly assigned to either experimental (BMO, *n* = 13) or control groups (CON, *n* = 13) and received one of the synthetic diets for two weeks (Figure 1).

### 2.3. Ethics Statement

The experiment was approved by the Animal Experimentation Committee under the Danish Food Administration and was licensed according to the Danish Animal Experimentation Act and the EU Directive 2010/63/EU (License No 2017-15-0201-01262).

### 2.4. Infant Type Co-Culture for Inoculation

The infant type co-culture was composed of eight bacterial type strains: *Bifidobacterium longum* subsp. *longum* (DSM 20219), *Bifidobacterium breve* (DSM 20213), *Lacticaseibacillus rhamnosus* (LMG 18243), *Staphylococcus aureus* (DSM 20231), *Staphylococcus epidermidis* (DSM 20044), *Clostridium perfringens* (DSM 756), *Parabacteroides distasonis* (DSM 20701) and *Escherichia coli* (DSM 30083). The eight bacteria were activated from frozen stocks into fresh optimal media as described previously [9] and 200 µL aliquots of co-culture containing approximately 10^8^ CFU of each strain were kept frozen until use. The co-culture was administered to the mice by holding them loosely in the neck skin and administering the entire content of one aliquot to the mouth using a pipette.

### 2.5. Sample Collection

Fecal samples were collected from the mice three days after weaning, halfway through the second week of intervention and on the day of sacrifice. In addition, fecal samples were collected from the cages before inoculation and on the day of sacrifice to check sterility. Consumption of the isolator diet and the experimental diets were registered for each cage weekly using a spring scale (Pesola Micro Line, Capacity = 30 g, Division = 0.25 g, Switzerland). Final body weight of the animals was registered on the day of sacrifice.

Mice were sacrificed and blood was sampled into EDTA-coated tubes and plasma obtained by centrifugation at 8000 g for 10 min. From the gastrointestinal tract, content of the cecum and proximal colon was sampled into individual tubes and snap frozen in liquid nitrogen. All samples were stored at −80 °C until further analysis.

### 2.6. Sample Preparation for ^1^H Nuclear Magnetic Resonance (NMR) Spectroscopy

Plasma samples were thawed at room temperature and then filtered through prewashed 0.5 mL 10k Millipore centrifugal filter units (Amicon Ultra, Merck Millipore Ltd., Billerica, MA, USA) by centrifugation at 4 °C at 10,000g for 1 h. A volume of 150 µL plasma filtrate was transferred to a 5 mm NMR tube with 300 µL D_2_O (deuterium oxide, 99.9%, Cambridge Isotope Laboratories, Andover, MA, USA) and 100 µL D_2_O containing 0.05% (*w/v*) 3-(trimethylsilyl)-2,2,3,3-tetradeuteropropanoic acid (TSP, Sigma-Aldrich, St. Louis, MO, USA).

Cecal content samples were thawed at room temperature and ~100 mg sample material was weighed out. A volume of 800 µL distilled H_2_O was added to achieve an approximate W (sample): V (H_2_O) ratio of 1:9 for extraction of metabolites. The sample was vortexed for 10 s and centrifuged at 4 °C at 14,000× *g* for 5 min. The supernatant was filtered through 0.5 mL 10k Millipore centrifugal filter units by centrifugation at 4 °C at 14.000× *g* for 30 min and the filtering procedure repeated until a total volume of 400 µL of filtered supernatant could be obtained. The pooled supernatant was added to 5 mm NMR tube with 50 µL phosphate buffer (pH 7.4, 0.6 M) containing internal standard 10 mM DSS (3-(trimethylsilyl)-1-propanesulfonic acid-d_6_sodium salt (Sigma-Aldrich, St. Louis, MO, USA) and 200 µL D_2_O. The pH was measured using a pH meter fitted with a silver electrode (Radiometer, Copenhagen, Denmark) before addition of buffer.

Colon content samples were thawed at room temperature and 15–50 mg of sample material was weighed out. A volume of distilled H_2_O corresponding to a W (sample): V (H_2_O) ratio of 1:20 was added to the sample, it was vortexed twice for 10–20 s to dissolve the pellet and centrifuged at 1000× *g* for 1 min. The supernatant was filtered through 10k Millipore centrifugal filter units at 10,000× *g* for 30 min and repeated until a volume of 400 µL filtrate was obtained. The collected supernatant was added to 5 mm NMR tubes with 100 µL phosphate buffer (pH 7.4, 0.6 M) containing 10 mM DSS and 100 µL D_2_O.

### 2.7. ^1^H NMR Spectroscopy and Metabolomics

For plasma, cecal and colon content samples, NMR spectra were acquired on a Bruker Avance III 600 MHz NMR spectrometer (Bruker BioSpin, Gmbh, Rheinstetten, Germany) operating at a proton NMR frequency of 600.13 MHz for ^1^H and equipped with a 5 mm TXI probe. The 1D NOESY pulse experiment with presaturation of the spectral region containing the water peak (noesypr1d) was used with a recycle delay of 5 s. A total of 128 FIDs were acquired and the acquisition parameters included 32K complex data points, a spectral width of 7289 Hz (12.15 ppm) and an acquisition time of 2.25 s. The measurements were performed at a temperature of 298K (25 °C). NMR spectra were automatically baseline corrected and manually phase corrected using Topspin (Version 3.0, Bruker BioSpin).

In Matlab (R2016, MathWorks Inc., Natick, MA, USA), processed NMR spectral files were aligned by icoshift [10] and calibrated to the TSP or DSS resonance at 0.0 ppm. Spectral regions with no or disturbing resonances were trimmed, by excluding the regions containing the ethanol resonances at 3.67–3.63 and 1.26–1.12 ppm, the water resonance at 4.90–4.75 ppm and finally the regions above 10.0 ppm and below 0.5 ppm. Finally, the area of the NMR spectra were normalized to the area of TSP or DSS and binned into 0.005 ppm intervals in order to reduce the amount of data points.

For multivariate analysis the pre-processed data were loaded into SIMCA (15, Sartorius Stedim Data Analysis AB), pareto-scaled and principal component analysis (PCA) performed to visualize the variation in the data. If PCA models showed clear separation between diet groups, orthogonal, partial least squares discriminative analysis (OPLS-DA) was performed using diet as Y-variable to maximize the separation between dietary groups and S-line plots were calculated to identify the bins contributing the most to group discrimination. Full cross-validation was performed using the leave-one-out method to estimate model validity (Q^2^). Metabolite identification and quantification was performed in Chenomx NMR suite (Chenomx Inc., Edmonton, Alberta, Canada) using the build-in 600 MHz library.

### 2.8. DNA Extraction

DNA was extracted from cecal content and fecal samples using the Bead-Beat Micro AX Gravity kit with mechanical lysis (A&A Biotechnology, Gdańsk, Poland) according to the manufacturer’s protocol with a few modifications. Briefly, cecal or fecal material was added to the bead-beat tubes, approximately 100 mg per tube. A reaction mixture containing 1 mL LSU buffer, 10 µL lysozyme and 5 µL mutanolysin was added to each tube, followed by vortexing and then incubation at 50 °C for 20 min. A volume of 20 µL Proteinase K was then added, followed by bead beating in a FastPrep-24 5G instrument (MP Biomedicals, Santa Ana, CA, USA) in three cycles of 20 s at a speed of 6.5 M/s. The samples were then incubated at 50 °C for 20 min and the remaining steps performed according to the manufacturer’s protocol. A NanoDrop spectrometer (Thermo Fisher Scientific, Wilmington, DE, USA) was used to determine the yield and purity of DNA. The yield was ranging 29–181 ng/µL and purity (A_260/280_) was 1.8–2.1 for all samples. After extraction, the DNA was stored at −60 °C until further processing.

### 2.9. High-Throughput 16S rRNA Gene Amplicon Sequencing

High throughput 16S rRNA gene amplicon sequencing was conducted on DNA from cecum content and fecal samples that were collected during the experimental period and at sacrifice to determine the microbiota composition. The primers, NXt_388_F (5′-TCGTCGGCAGCGTCAGATGTGTATAAGAGACAGACWCCTACGGGWGGCAGCAG-3′) and NXt_518_R (5′-GTCTCGTGGGCTCGGAGATGTGTATAAGAGACAGATTACC GCGGCTGCTGG-3′) were designed with adapters Nextera Index Kit^®^ (Illumina, CA, USA) and targeted the V3 region (~190 bp). Amplicon library preparation, purification and sequencing was performed similar as described by Castro-Mejía [11]. Briefly, purification of the amplified fragments was performed by AMPure XP Beads (Beckman Coulter Genomic, CA, USA) and prior to library pooling, the clean constructs were quantified using a Qubit Flourometer (Invitrogen, Carslbad, CA, USA) and mixed in approximately equal concentrations to ensure even representation of reads per sample. Finally, 2 × 150 cycles, pair-ended Illumina NextSeq sequencing using the Mid Output v2 chemistry (Illumina, CA, USA) was performed. High-throughput sequencing data were processed as previously described [11]. Briefly, the raw data containing pair-ended reads were merged and trimmed as described previously [12]. Zero radius operational taxonomic units (zOTUs) were conducted using the UNOISE pipeline [13]. The green genes (v13.8) 16S rRNA gene collection was used as a reference database [14]. Reads from OTU tables were aggregated on the family level, which was the most robust level of discriminating the bacteria in the infant type coculture. Information on genus was investigated if relevant to the interpretation of results. Sequences assigned to OTU f__Streptococcaeceae;g__Lactococcus were filtered out as they originated from Lactococcus precipitated casein in the diet [15]. Samples with low DNA concentration (<0.1 ng/µL) were excluded from the abundance analysis. The R software (version 3.6.1, The R Foundation, Vienna, Austria) was used for calculating and visualizing the relative abundances constituting more than 2% of the total relative abundance.

### 2.10. Real Time Quantitative Polymerase Chain Reaction (RT qPCR)

A SYBR™ Green assay using primers designed to target the non-conserved regions of the 16S rRNA gene of selected bacterial species was set-up. The primers are described elsewhere [9]. The bacterial DNA from pure cultures was serially diluted in RNase free water (10^−1^ to 10^−6^), while DNA from cecal samples was analyzed without dilution. One master mix per primer set was prepared consisting of Fast SYBR^®^ Green I Master Mix, primer mix (forward and reverse diluted to 5 μM) and RNAse free water. Five μL DNA or RNAse free water for non-template controls (NTC) was added to reach a final reaction volume of 20 μL. Two technical replicates were included per biological sample. The RT qPCR assay was run on a 7500 Fast Real-Time PCR System (Life Technologies™, Dublin, Ireland) using the following program: 50 °C (2 min), at 95 °C (2 min), forty cycles of: 95 °C (15 s), 55 °C (15 s) and 72 °C (1 min). Melting curve analysis was then run at 95 °C (15 s), 56 °C (1 min), 95 °C (30 s) and 56 °C (15 s) to determine the melt temperature of the amplicon.

### 2.11. Statistical Analysis

ANOVA was applied to test the effect of treatment group (BMO or CON diet) on feed intake, weight of mice at the end of intervention as well as treatment effect on metabolite concentrations in cecum content, colon content and plasma. Before ANOVA analysis, the datasets were checked for variance homogeneity and normality by visual inspection of residual vs. fitted and QQplots, respectively. Deviating samples were only removed if differed substantially from the rest of the samples of the same treatment, or if the sample contributed to non-normality of the dataset and an underlying reason for removal of the sample could be identified. Data analysis and visualization was performed in R (version 3.6.1, The R Foundation, Vienna, Austria).

## 3. Results

During the experimental period, three mice died, one of which were fed the BMO diet and two who were fed the CON diet. Unfortunately, additional mice showed contamination of fecal or cecum samples and were therefore excluded from all analyses. The final distribution of mice on each diet were BMO (*n* = 11) and CON (*n* = 9). Consumption of the BMO diet was higher than CON diet at week 1 (Figure 2A, *p* = 0.037) and in total over the two weeks (Figure 2C, *p* = 0.028), but the final body weight of the mice in each treatment group was not significantly different (Figure 2D, *p* = 0.83).

### 3.1. Microbiota Composition during Experimental Period

Only two of the eight strains in the infant type coculture were identified on the family level. The microbiota composition during the experimental period (Figure 3) was dominated by the families *Enterobacteriaceae* and *Lactobacilliaceae*, which correspond to the families of *E. coli* and *L. rhamnosus* of the infant type co-culture, respectively. Collectively, the data did not indicate differential microbiota response between the two treatments after 7 days of intervention; the average relative abundance of *Enterobacteriaceae* was 89% (±6%) and 86% (±10%) and *Lactobacilliaceae* constituted 11 (±6%) and 14% (±10%) in BMO and CON treatments, respectively. Intriguingly, the other bacterial families that constituted the infant type co-culture were not observed in these fecal samples taken 7 days after inoculation.

### 3.2. Microbiota Composition and Metabolites in Colon

Due to limited sample material, it was not possible to compare the microbiota composition and metabolite profile of colon content one-to-one. Therefore, microbiome composition was performed on freshly collected fecal samples from each individual mouse before sacrifice and NMR metabolomics analyses were performed on proximal colon content after sacrifice. The fecal samples obtained at sacrifice showed a high abundance of *Enterobacteriaceae* and minor abundance of *Lactobacilliaceae* (Figure 3, mid panel), while other families constituting the infant type co-culture were not detected. Specifically, the relative abundance of *Enterobacteriaceae* was 97% (±4%) in the BMO group and 98% (±3%) in the CON group, while *Lactobacilliaceae* constituted 2% (±4%) in the BMO group and 2% (±3%) in the CON group. NMR metabolomics analysis showed a clear separation between the two treatment groups and a valid OPLS-DA model was obtained (Figure 4A, Q2 = 0.903). Resonances that were enhanced with the BMO treatment included 3′-SL (the main sialyllactose in the BMO product), 4-hydroxyphenyl acetate and other BMO resonances around 8.5 ppm, while the resonance from succinate was enhanced with the CON treatment. Examination of the NMR spectra did not reveal resonances from lactose in colon content of mice from either of the dietary treatments (data not shown).

Univariate statistical analyses revealed that the concentration of formate was higher in the CON treatment (*p* = 0.05) and so was succinate (*p* = 0.019), but the concentrations of SCFAs and other organic acids were not significantly different between the two treatment groups (Figure 5A).

### 3.3. Microbiota Composition and Metabolites in Cecum Content

Cecum content was sampled after sacrifice and divided into two aliquots: one for DNA extraction (16S rRNA gene amplicon sequencing and RT qPCR) and one for 1H NMR metabolomics. This allowed for a direct comparison between microbiota composition and metabolic pattern. The microbiota data (Figure 3, lower panel) showed that the cecum content was dominated by *Enterobacteriaceae*. The relative abundance of *Enterobacteriaceae* was above 99% in both BMO and CON treatments. *Lactobacilliaceae* were also present, but the relative abundance was lower than the 2% cut-off value. RT qPCR was performed to determine the absolute quantity of the two strains using primers specific for *L. rhamnosus* and *E. coli*. The results showed that *L. rhamnosus* was not significantly different between cecum samples from BMO and CON treatments (1.8 ± 1.3 and 2.3 ± 0.8 logCFU/g cecum content, *p* = 0.2). The absolute quantity of *E. coli*, however, was significantly lower in BMO (7.9 ± 1.3 logCFU/g cecum content) compared to CON (9.5 ± 2.1 logCFU/g cecum content) treatment (*p* = 0.036). Multivariate analysis of the ^1^H NMR spectra showed a clear separation between the two dietary groups when employing OPLS-DA (Figure 4B, Q2 = 0.918). Resonances that were positively associated with BMO treatment were assigned to 3′SL, 4-hydroxyphenyl acetate and other BMO resonances at 8.0 ppm. The CON treatment was associated with higher intensities of succinate and an unassigned multiplet at 1.85 ppm. Examination of the NMR spectra did not reveal resonances from lactose in cecum of mice from either of the dietary treatments (data not shown). Succinate was significantly higher in the CON treatment (*p* = 0.013), while isobutyrate tended to be higher (*p* = 0.061). The remaining SCFA and other organic acids did not reach statistical significance (Figure 5B).

### 3.4. Relative Abundance of Enterobacteriaceae and Lactobacilliaceae at Weaning, Experimental Period and Sacrifice

The abundance of *Enterobacteriaceae* and *Lactobacilliaceae* showed major variation at different time points and in different sample types (colon or cecum) in the study (Figure 6). The abundance of *Lactobacilliaceae* peaked in feces during the experimental period, decreased slightly in feces collected at sacrifice and was only present in very low abundance in cecum content collected at sacrifice (Figure 5A). *Enterobacteriaceae* increased after weaning to a point where this family constituted almost the entire population in cecum at sacrifice (Figure 6A).

### 3.5. Metabolites in Plasma

Acetate, butyrate, formate and lactate were quantified in the blood plasma samples collected on the day of sacrifice (Figure 5C). High concentrations of lactate were found in both dietary groups. None of the metabolites were significantly different between the two dietary groups.

## 4. Discussion

In the present study, germ-free weaning mice were inoculated with an infant type co-culture to produce a gnotobiotic mouse model for studying the in vivo effect of BMO supplementation on microbiota composition, metabolic activity in the lower gastrointestinal tract and alterations in plasma metabolites. Gnotobiotic mouse studies allow for the investigation of in vivo effects of, e.g., prebiotics in a human-like microbiota and provide unique prospects for confirming associations and proving causality in microbiota research [16]. However, a key challenge with gnotobiotic mice studies relates to the successful colonization of animals with a microbiota that is not co-evolved with the host.

In the present study, feeding of either a BMO diet containing 2% bovine milk oligosaccharides or a control (CON) diet with lactose for two weeks resulted in fecal samples and a cecum content highly dominated by *Enterobacteriaceae* and *Lactobacilliaceae*. In contrast, none of the other strains included in the infant type co-culture were detected using the high-throughput 16S rRNA gene amplicon sequencing approach. A previous study with infant type co-culture inoculation showed successful colonization of germ-free mice after administering a single gavage of infant type co-culture [17]. Following the infant type co-culture inoculation, the mice were fed a natural ingredient chow diet for four weeks and at the end of the experiment, the mice had a microbiota dominated by *Enterobacteria*, *Bacteroides* and *Clostridium* [17]. Interestingly, Martin et al. (2007) also identified bifidobacteria in the feces at the end of the experiment, which is in contrast, to the present study. Although the mode of inoculation was similar to Martin et al. (2007), the diets were different. In the present study, synthetic diets were formulated with a relatively high content of lactose (14/100 g). It is very likely that the high lactose content of the diets in the present study favoured the growth of *E. coli* and *L. rhamnosus* and thus, colonization by *Bifidobacteriaceae* and other species of the infant type co-culture was not accomplished. Since *Bifidobacteriaceae* is expected to be decisive for the metabolic activity of BMO-supplementation, the study did not allow for examination of the anticipated effects. Early studies of humanized mice have shown shifts in the dominant species of microbiota after administration of microbiota from humans [18]. Furthermore, several studies have shown that successful colonization of germ-free mice with a human microbiota is challenging and especially colonization by human co-evolved *Bifidobacterium* species is not always successful. While mono-association of germ-free mice with the human co-evolved *Bifidobacterium bifidum* resulted in stable colonization, this strain was outcompeted by murine microbiota, when other mouse co-evolved bacteria were present [19]. The murine microbiota contained *Bifidobacterium animalis*. Cell assays further showed that *B. animalis* showed stronger adhesion to murine intestinal epithelial cells than to human intestinal cells, which might be the underlying mechanism for outcompeting the *B. bifidum* strain [19]. Another study by Charbonneau et al. (2016), showed that administration of sialylated BMO to gnotobiotic mice with a Malawian infant type microbiota similarly did not show colonization by *Bifidobacterium* [7]. The overgrowth of *Enterobacteriaceae* observed in the current study is probably a result of a low host association of the remaining bacterial strains in the infant type co-culture to the mouse gastrointestinal tract, leading to overgrowth of *Enterobacteriaceae*. *E. coli* strains such as the one used in the infant type co-culture are fast growing bacteria with a generation time of 20 min at optimal conditions [20]. Therefore, it is likely that this strain outcompeted most of the other bacterial strains in the infant co-culture and therefore the possibility of observing any differences between treatment groups was impeded by *E. coli* overgrowth cecum and colon. The significant ~1.5 logCFU/g higher *E. coli* numbers identified in cecum content samples from the CON treatment is intriguing. Previous results from our lab showed inhibition of *E. coli* growth in infant type coculture when BMO was present in the media together with lactose compared to media that only contained lactose [9]. The in vitro results further showed higher *B. longum* subsp. *longum* growth in the treatments with BMO and lactose, while *L. rhamnosus* growth was not detected. This is in contrast to the present study, where *Bifidobacterium* were not detected in cecum or fecal samples. These differences between in vitro and in vivo studies might be due to the evolutionary adaptation of *Lactobacillus* to colonize the gastrointestinal tract of mice. *Lactobacilli* (family *Lactobacilliaceae*) are consistently found in high relative abundance in murine microbiota, whereas *Bifidobacteria* (family *Bifidobacteriaceae*) are only present in low relative abundance [21]. This suggests that *lactobacilli* are more likely to colonize the gastrointestinal tract of mice than bifidobacteria, supporting the finding in the present study that *Lactobacilliaceae* were identified in feces and cecum content, while *Bifidobacteriaceae* were not.

Previous studies have shown that cecum content and feces share great similarities in microbial populations, but cecum content shows better separation between treatment groups than end-point fecal samples [22]. Cecum is also the site of the gastrointestinal tract in mice, where the majority of fermentation processes occur [16]. The results from the present study show differences but also similarities in microbiota composition in cecum and fecal samples collected on the day of sacrifice. Specifically, the relative abundance of *Lactobacilliaceae* appears to be slightly higher in fecal samples, as compared to cecum samples.

In the present study, the metabolic profiles of cecum content and colon content were very similar. Lactose was not detected in the cecum or colon content from mice in either treatment group, indicating that lactose in the diet is either absorbed in the small intestine of the mouse before the dietary bolus reaches the cecum or efficiently fermented by *E. coli* and L. rhamnosus in the cecum. In contrast, un-metabolized dietary components, such as 3′-SL, was identified in cecum and colon content from samples the BMO treatment. This finding reveals that 3′-SL was not degraded during passage of the upper gastrointestinal tract and reached the lower gastrointestinal tract where it was available for bacterial fermentation. We speculate whether the presence of 3′SL and other BMO structures in the cecum and colon of the mice might have had effects in the gastrointestinal tract such as spatial organization of the Enterobacteriaceae, mucus thickness or other parameters related to the gastrointestinal tract, which we did not measure in the present study.

## 5. Conclusions

The present study describes how the application of NMR-based metabolomics in combination with high-throughput 16S rRNA gene amplicon sequencing was useful in comparing metabolic activity, microbiota composition in different compartments of the lower gastrointestinal tract using a gnotobiotic mice model. BMO components were detected in cecum and colon contents, but did not result in microbiota alterations, except for a slightly higher *E. coli* population in cecum content from CON treatment and slightly higher formate and succinate concentrations in gastrointestinal samples. The unsuccessful colonization by infant type microbiota warrants more studies into how gnotobiotic mouse models can be optimized for studying human microbiota.

## Figures and Tables

**Figure 1 microorganisms-09-01003-f001:**
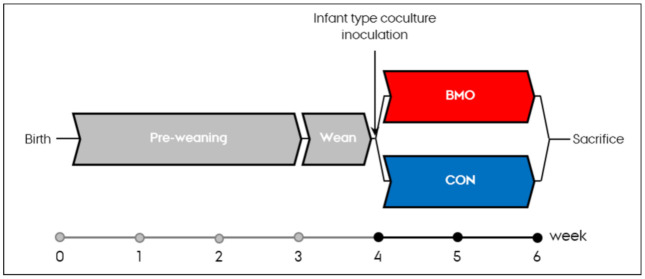
Study design of the gnotobiotic mouse study showing the time from birth, pre-weaning, weaning, experimental period until sacrifice. BMO: synthetic diet containing 2% (*w/w*) bovine milk oligosaccharides. CON: synthetic control diet.

**Figure 2 microorganisms-09-01003-f002:**
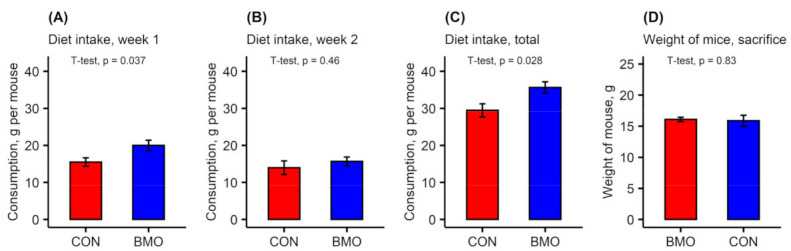
Background data on mice. (**A**–**C**) Diet consumption during week 1 and 2 and in total over the course of the experimental period was measured per cage and was normalized to the number of animals in individual cages to express g/mouse and (**D**) Body weight of mice at the end of the experimental period (BMO: *n* = 11, CON: *n* = 9). Values are means and error bars show standard deviation.

**Figure 3 microorganisms-09-01003-f003:**
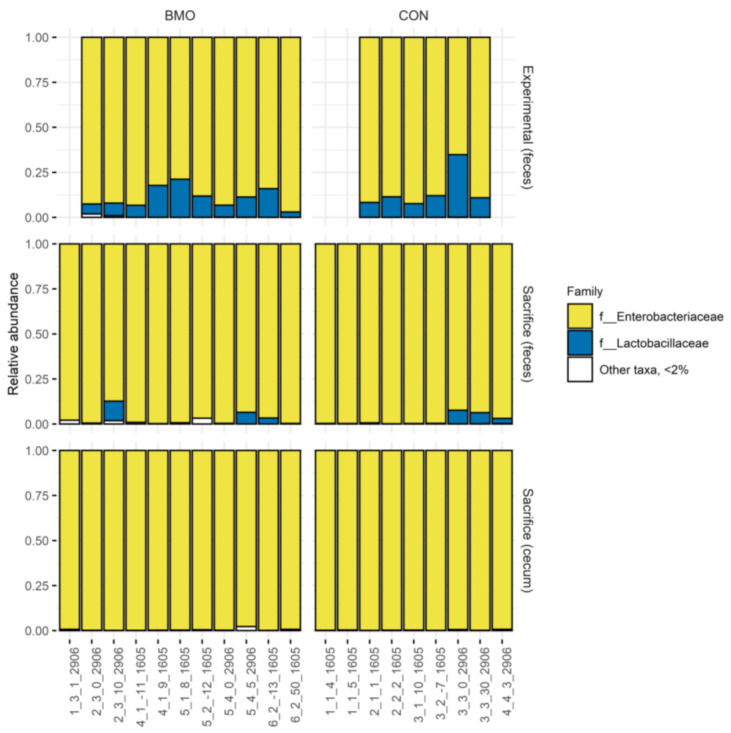
Relative abundance at family level (<2% relative abundance), horizontally grouped according to dietary treatment and vertically grouped according to sampling time (experimental or sacrifice) and sample type (feces or cecum content). Each vertical bar with a unique number represents samples from an individual mouse. Sequencing data of fecal samples from the experimental period was available from a total of 16 mice (BMO; *n* = 10, CON: *n* = 6), while for at the time of sacrifice, sequence data were available fecal and cecum content for a total of 20 mice (BMO: *n* = 11, CON: *n* = 9) BMO: synthetic diet with 2% bovine milk oligosaccharides, CON: synthetic control diet with lactose.

**Figure 4 microorganisms-09-01003-f004:**
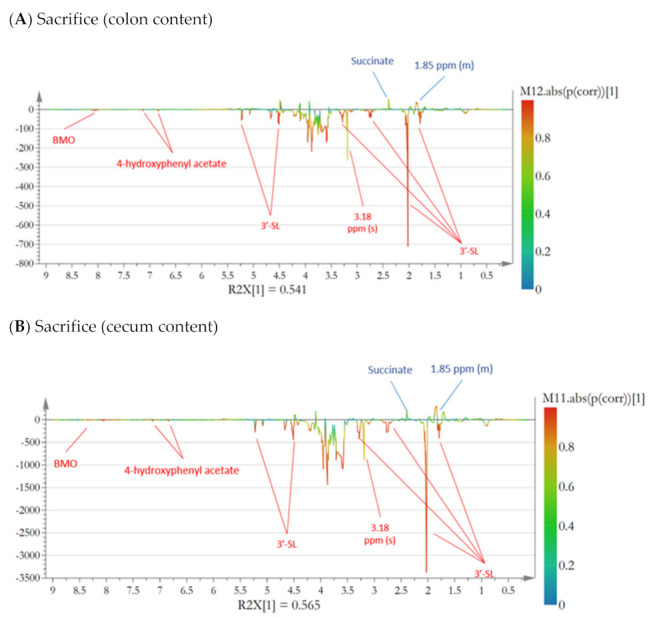
S-line plot from OPLS-DA showing ^1^H NMR metabolomics data from samples collected at sacrifice. Positive resonances are correlated to CON treatment and negative resonances are correlated to BMO treatment. Color of the line indicates the significance of correlation. The OPLS-DA model validated using full-cross validation (leave-one-out) and each included one predictive and one orthogonal component, (**A**) Colon content, (Q2-value = 0.903), (**B**) Cecum content, (Q2 = 0.918).

**Figure 5 microorganisms-09-01003-f005:**
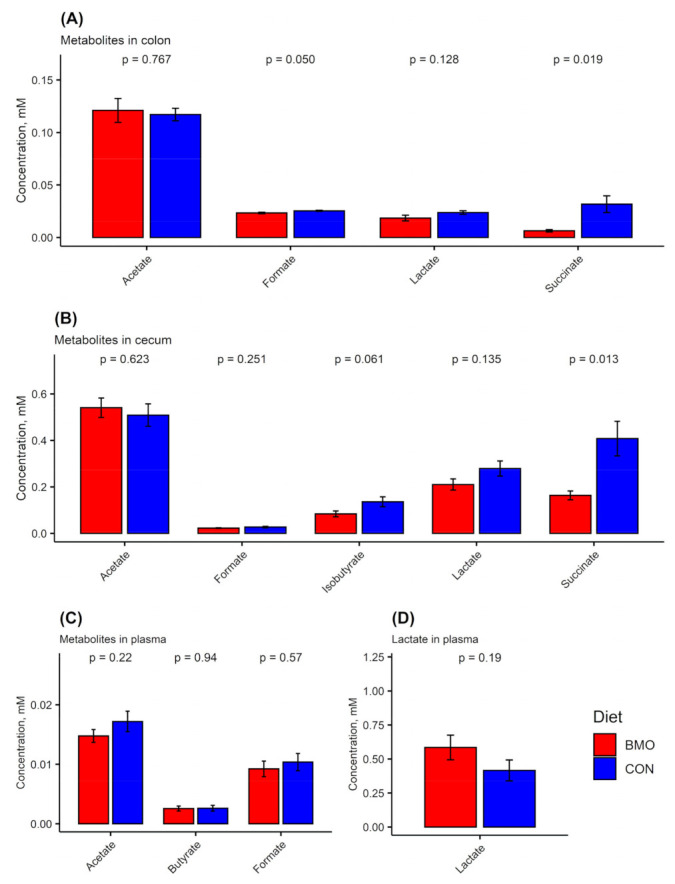
Quantified metabolites at sacrifice in (**A**) colon content (BMO: *n* = 7, CON: *n* = 7), (**B**) cecum content (BMO: *n* = 11, CON: *n* = 8), (**C**) metabolites in plasma and (**D**) lactate in plasma (BMO: *n* = 11, CON: *n* = 8). *p*-values indicate significance level t-test on effect of diet on metabolite concentration. BMO: diet with 2% bovine milk oliogsaccharides, CON: control diet with lactose. Values are means and error bars show standard deviation.

**Figure 6 microorganisms-09-01003-f006:**
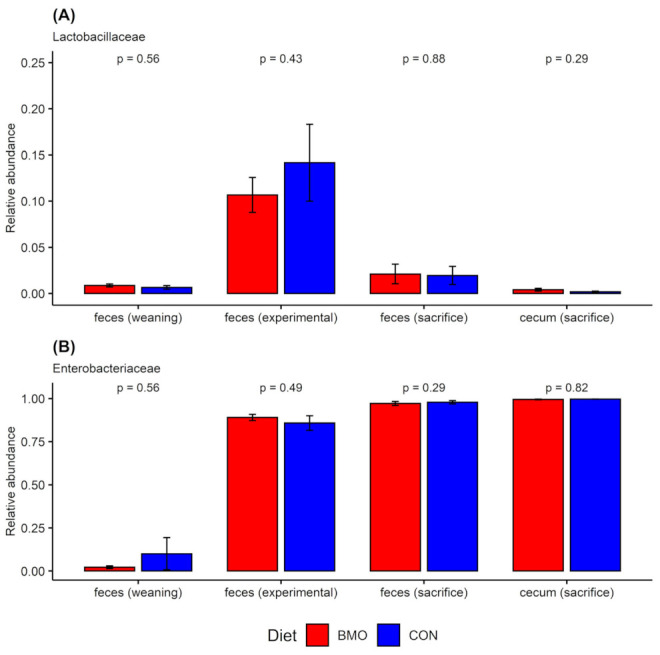
Relative abundance of selected families in samples collected throughout the study. (**A**) Lactobacilliaceae, (**B**) Enterobacilliaceae. The x-labels indicate the sample type and time of collection: feces (weaning): feces collected during weaning (BMO *n* = 5, CON *n* = 4), feces (experimental): feces collected midway through the experimental period (BMO *n* = 10, CON *n* = 6), feces (sacrifice): feces collected before sacrifice (BMO *n* = 11, CON *n* = 9), cecum (sacrifice): cecum content collected after sacrifice (BMO *n* = 11, CON *n* = 9). Colors indicate the dietary treatment: BMO: synthetic diet with 2% bovine milk oligosaccharide, CON: control diet with lactose. Dots show individual samples, bars are mean values and error bars show standard error. *p*-values above bars are *t*-test comparing the average relative abundance in samples from the two diets collected at the same time point.

**Table 1 microorganisms-09-01003-t001:** Nutritional composition of the BMO and CON diet.

Per 100 g	BMO	CON
Energy [Kcal]	351	367
Protein [g]	15.8	16.6
Carbohydrate [g]	57.5	60.2
Lactose [g]	13.6	14.3
Fat [g]	6.4	6.7
Fibre [g]	8.8	9.3
Oligosaccharide [g]	2.0	0.0

## Data Availability

Data can be released upon request to the corresponding author.

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
