# Peer review of "Administration of Bovine Milk Oligosaccharide to Weaning Gnotobiotic Mice Inoculated with a Simplified Infant Type Microbiota"

_microorganisms, 2021, doi:10.3390/microorganisms9051003_

Round 1

Reviewer 1 Report

Nice paper on an animal model showing the microbiological effects of bovine milk oligosaccharides.

The English used is adequate and just needs a fair revision by authors. 

It would be nice adding the p value to table 1.

Why in Figure 2 has been used the ANOVA as statistical test ? is it a t-test or chi-square ?

Reviewer 2 Report

In this manuscript, authors examined the effect of Bovine milk oligosaccharides (BMO) in gnotobiotic mice. I found that the manuscript has important weaknesses that need to be improved. One of these is the low number of (n) for certain analysis, which limits certain conclusions. Please also see my other comments below.

Major comments:

Lines 105-106: Improve the quality of the graphs. Same comment for all graphs in the manuscript.

Lines 274-275: In the chart legend, the authors wrote n = 11 for the BMO group and n = 9 for the CON group. However, for graphs A, B and C the points do not show the same number as in the legend. Please clarify this point.

Lines 279-280: For the gut microbiota analysis why are the authors showing the results only at the family level and haven't gone deeper at the taxonomic level? Please clarify this point. Did they analyze the richness? Same comment for the graphs as above.

Line 287: In this section, the bacteria families also go to Italique.

Lines 360-361: For the metabolite analysis, I have found some rather low values and for some of them there seems to be no detection. Was some metabolites too low due to an analytical problem? Did the authors use any internal standards to perform the analysis? I also found that the n is low enough to draw some conclusions about it.

Round 2

Reviewer 2 Report

I really appreciate all the changes done by the authors. I do not have neither minor nor major comments.